# The Landscape and Therapeutic Implications of Molecular Profiles in Epithelial Ovarian Cancer

**DOI:** 10.3390/jcm9072239

**Published:** 2020-07-15

**Authors:** Ludivine Dion, Isis Carton, Sylvie Jaillard, Krystel Nyangoh Timoh, Sébastien Henno, Hugo Sardain, Fabrice Foucher, Jean Levêque, Thibault de la Motte Rouge, Susie Brousse, Vincent Lavoué

**Affiliations:** 1Service de Chirurgie gynécologique, CHU de Rennes, 35000 Rennes, France; ludivine.dion@chu-rennes.fr (L.D.); isis.carton@hotmail.fr (I.C.); krystel.nyangoh.timoh@chu-rennes.fr (K.N.T.); hugo.sardain@chu-rennes.fr (H.S.); fabrice.foucher@chu-rennes.fr (F.F.); jean.leveque@chu-rennes.fr (J.L.); Susie.brousse@chu-rennes.fr (S.B.); 2Faculté de médecine, Université de Rennes 1, 35000 Rennes, France; sylvie.jaillard@chu-rennes.fr; 3INSERM U 1085, IRSET, Equipe 8, 35000 Rennes, France; 4Service de Cytogénétique, CHU de Rennes, 35000 Rennes, France; 5Service d’anatomo-pathologie, CHU de Rennes, 35000 Rennes, France; sebastien.henno@chu-rennes.fr; 6Service d’oncologie médicale, CRLCC Eugène Marquis, 35000 Rennes, France; t.delamotterouge@rennes.unicancer.fr

**Keywords:** ovarian cancer, molecular subtypes, BRCA, BRCAness profile, therapeutic approach

## Abstract

Epithelial ovarian cancer (EOC) affects 43,000 women worldwide every year and has a five-year survival rate of 30%. Mainstay treatment is extensive surgery and chemotherapy. Outcomes could be improved by molecular profiling. We conducted a review of the literature to identify relevant publications on molecular and genetic alterations in EOC. Approximately 15% of all EOCs are due to *BRCA1* or *BRCA2* mutations. Four histologic subtypes characterized by different mutations have been described: serous, endometrioid, mucinous, and clear-cell. Between 20–30% of high-grade serous EOCs have a *BRCA* mutation. Tumors with *BRCA* mutations are unable to repair double-strand DNA breaks, making them more sensitive to platinum-based chemotherapy and to PolyAdenosine Diphosphate-Ribose Polymerase (PARP) inhibitors. Olaparib is a PARP inhibitor with proven efficacy in *BRCA*-mutated ovarian cancer, but its effectiveness remains to be demonstrated in tumors with a BRCAness (breast cancer) profile (i.e., also including sporadic tumors in patients with deficient DNA repair genes). A universally accepted molecular definition of BRCAness is required to identify optimal theranostic strategies involving PARP inhibitors. Gene expression analyses have led to the identification of four subgroups of high-grade serous EOC: mesenchymal, proliferative, differentiated, and immunoreactive. These subtypes are not mutually exclusive but are correlated with prognosis. They are not yet used in routine clinical practice. A greater understanding of EOC subtypes could improve patient management.

## 1. Introduction

Epithelial ovarian cancer (EOC) affects one in every 60 women in industrialized countries [1] and has an annual incidence of approximately 43,000 cases worldwide [2]. It mainly affects women after menopause. The prognosis for ovarian cancer is poor, with a five-year survival rate of just 30% and most deaths occurring within two years of diagnosis. Approximately 75% of all women with EOC are diagnosed when the cancer has spread from the ovaries to the entire peritoneal surface (International Federation of Gynecology (FIGO) stages IIIc and IV). The current five-year survival rate for patients with FIGO stage IIIc–IV EOC is less than 30% and no real improvements have been achieved in recent decades [1]. Surgery combined with carboplatin and paclitaxel chemotherapy was the current standard of care prior to the recent development of antiangiogenic agent bevacizumab, which resulted in a modest improvement in progression-free survival (four months) in women with ovarian cancer with a poor prognosis, i.e., women with stages other than FIGO stage IIIc with residual disease after surgery and FIGO stage IV [3]. No significant improvements were observed for overall survival. Olaparib has also provided clear survival benefits in patients with BReast CAncer *(BRCA) 1* and *BRCA2* mutations [4]. A better understanding of the mechanisms underlying treatment failure and the identification of new therapeutic targets are needed in EOC. In this review, we provide an updated overview of the different molecular profiles described for EOC in the literature and discuss their role in the current and future management of this disease.

## 2. Materials and Methods

We performed a literature search of the PubMed database to identify studies relating to molecular and genetic alterations in ovarian cancer in humans. The search was based on the keywords “epithelial ovarian cancer”, “mutation”, “PolyAdenosine Diphosphate-Ribose Polymerase (PARP) inhibitor”, “molecular profiles”, and “treatments” and targeted original research articles, case reports, meta-analyses, and reviews published in English between January 2000 and January 2020.

Identification of duplicate studies was facilitated by downloading all the key fields for each study, including unique identifier numbers (e.g., PubMed identifiers (PMIDs) and digital object identifiers (DOIs)), clinical trial numbers, abstracts, and keywords. Titles and abstracts of the studies retrieved were screened for relevance using predefined inclusion and exclusion criteria. Relevant studies were selected for full-text review and data extraction; all other studies were excluded. A search of the gray literature was also conducted to identify unpublished journal articles and conference proceedings, and clinical trial registers were searched for unpublished and ongoing trials. Finally, a hand search was made of the reference lists of selected publications to identify other potentially relevant studies.

## 3. Results

The literature search identified 1079 articles relating to the molecular classification of ovarian cancer; 52 met the inclusion criteria and were included in this review.

### 3.1. The Natural History of EOC and Genetic Predisposition

The natural history of ovarian cancer is marked by a good initial response to standard treatment (surgery and chemotherapy) in 80% of women. Nonetheless, 70% of these women will experience disease recurrence, mostly in the form of diffuse peritoneal carcinomatosis, within two years of treatment completion. EOC is classified as platinum-resistant when disease recurs within six months of the end of platinum-based chemotherapy and as platinum-sensitive when it recurs after 12 months of treatment completion, although the period used to define sensitivity varies in the literature [5]. Tumors that recur between these periods are considered to have intermediate sensitivity.

Between 10% and 15% of women with EOC are genetically predisposed to ovarian cancer, and 90% have a *BRCA* mutation (*BRCA1* or *BRCA2*). The lifetime risk of EOC is 40% in patients with a *BRCA1* mutation versus 15% in those with a *BRCA2* mutation. Hereditary nonpolyposis colorectal cancer (Lynch syndrome) accounts for approximately 10% of all hereditary EOCs. Less common genetic syndromes, such as Li–Fraumeni syndrome, caused by a *P53* mutation, also exist [6,7,8,9,10]. In a study of inherited mutations in women with ovarian cancer, Norquist et al. [11] found that 18.1% of women had germline mutations in cancer-associated genes. Overall, 14.6% had mutations in *BRCA1* or *BRCA2*, 3.3% had mutations in other BRCA–Fanconi anemia genes (*BRIP1*, *PALB2*, *RAD51C*, *RAD51D*, *BARD1*), and 0.4% had mutations in mismatch repair genes linked to Lynch syndrome [10,11].

### 3.2. Histologic Subtypes of Ovarian Cancer, Molecular Correlates, and Therapeutic Implications

Four histologic subtypes of EOC have been identified: serous EOC (which is classified as high grade (71% of all EOCs) or low grade (4.1%)), endometrioid EOC (8.3%), clear-cell EOC (9.5%), and mucinous EOC (3.2%) [12]. However, observation of histological findings of tubal carcinoma in organs removed prophylactically from women with a genetic predisposition to ovarian cancer [13] led to the establishment of a dualistic classification system that divided EOC into type I (low-grade) carcinoma and type II (high-grade) carcinoma (Table 1).

Type I EOCs account for 25% of all EOCs, but cause just 10% of deaths; they are largely diagnosed at an early stage and have four histologic subtypes characterized by different mutations: (1) low-grade serous EOC (*KRAS*, *BRAF*, *HER2*), (2) endometrioid EOC (*PTEN* and *PI3KCA*), (3) mucinous EOC (*KRAS*), and (4) clear-cell EOC (*PI3KCA*). They have high genomic stability and span a continuum ranging from a benign cyst to a borderline lesion to an invasive tumor. Type II EOCs are much more common and lethal. They account for 75% of all EOCs and are responsible for 90% of deaths. They are typically diagnosed at an advanced stage (carcinomatosis), progress quickly, and include high-grade serous EOC, high-grade endometrioid EOC, carcinosarcoma, and undifferentiated carcinoma. Ninety-six percent of type II EOCs have a *P53* mutation, while between 30% and 50% have a *BRCA1* or *BRCA2* mutation (the percentage varies depending on the application of BRCA or BRCAness criteria) [14]. The concept of ‘BRCAness’ defines the pathogenesis and vulnerability of multiple cancers. The canonical definition of BRCAness is a defect in homologous recombination repair, mimicking BRCA1 or BRCA2 loss. In turn, BRCA-deficient cells utilize error-prone DNA repair pathways, causing increased genomic instability, which may be responsible for their sensitivity to DNA damaging agents and poly(ADPribose) polymerase (PARP) inhibitors. Other defects in homologous recombination repair genes, such as *EMSY*, *RAD51*, *ATM*, *ATR*, *Fanconi anemia*, *BARD1*, *BRIP1*, *PALB2*, *RB1*, *NF1*, *CDKN2A*, and the suppression of BRCA1 transcriptional activation through gene methylation, are associated with homologous recombination deficiency (HRD) [15,16]. The finding that HRD contributes to approximately 50% of type II EOC provided a rationale for using cytotoxic platinum-based chemotherapy and exploring the activity of PARP inhibitors in type II EOC. It has been demonstrated that PARP inhibitors are active in high grade serous EOC beyond those with BRCA mutations [17].

Type II EOCs have high genomic instability. Their precursor lesions are serous tubal intraepithelial carcinomas which are identical in appearance to high-grade serous papillary carcinomas and have the same *P53* mutations [13]. Other pathways in type II EOCs involved the overexpression of CCNE1, a gene encoding cell cycle protein cyclin E1, which leads to unscheduled DNA replication, centrosome amplification, and overall chromosomal instability [16] and also *FXM1*, which is altered in nearly 84% of high grade serous EOC, followed by Rb1 (67%), phosphatidylinositol-4,5-biphosphate 3-kinase (PI3K) (45%), and Notch 1 (22%) pathways, all of which could lead to novel therapeutic opportunities for patients with high grade serous EOC [10].

Although the dualistic classification of EOC into “type I” and “type II” is widely applied in research presentations and manuscripts, it is often used as a convenient way of conceptualizing different mechanisms of tumorigenesis amongst EOC. This classic dualistic classification conflicts with some recent molecular studies as not all type I EOCs are not homogenous, even within the histological types, and a proportion can have poor clinical outcomes such as high grade mucinous features because of the high rate of proliferation and advanced stage diagnosis (Table 2). Indeed, the impression that only type II carcinomas are aggressive, have poor prognosis, and carry TP53 mutations is not realistic. Although type II serous and type I low grade serous carcinomas best fit the description of the dualistic model, with different precursors and very distinct pathology and molecular profiles, there are very clear differences between mucinous ovarian carcinomas and other type I tumors. These include an unknown cell of origin and a very heterogeneous mutation profile (including TP53 mutant cases) and clinical behavior, indicating a non-type I classification for this entity. Thus, a greater understanding of the genomic mutations associated with the different histologic subtypes of EOC could help identify targeted systemic treatments.

Low-grade serous ovarian cancer represents about 10% of EOCs [18]. It is hypothesized that low-grade serous ovarian cancer develops from an atypical, proliferative (borderline) tumor [18]. Mutations in BRAF (38%) or KRAS (19%) are the most common aberrations detected in low grade serous EOC [19]. Although they are associated with a constitutively active Mitogen-Activated Protein Kinase (MAPK) pathway, mutations in BRAF or KRAS represent a favorable prognostic factor [18]. An activated MAPK pathway is detected in up to 80% of low-grade serous ovarian cancers and in 78% of their putative precursors (serous borderline tumors), suggesting a causative effect [10,20]. This point also provides a rationale for exploring MAPK kinase (MEK) inhibitors in the treatment of low-grade serous ovarian cancer, such as selumetinib, trametinib or dabrafenib, which are MEK1 and MEK2 inhibitors (MEKi) that indirectly inhibit the BRAF and KRAS pathways [21]. However, the MEKi selumetinib has shown little activity in patients with low-grade serous cancers (15%), where the response did not always correlate with KRAS/BRAF mutation status [22]. With another MEKi, clinical response to the BRAF inhibitor, dabrafenib, has been demonstrated thus far clinically in 20% women with BRAF V600E mutated low-grade carcinomas [23].

Clear-cell ovarian cancers display mutations in the SWItch/Sucrose Non-Fermentable (SWI/SNF) chromatin-remodeling complex genes, the PI3K/Akt signaling pathway, and the receptor tyrosine kinase (RTK)/Ras signaling pathway in nearly 50%, 40%, and 29% of clear-cell ovarian cancers, respectively [10,24,25]. Among the SWI/SNF subunits, ARID1A is the most frequently mutated gene, detected in 40% to 67% of clear-cell ovarian cancers [26,27]. In the PI3K signaling pathway, activating mutations in PIK3 catalytic subunit α (PIK3CA) (33%) and loss of phosphatase and tensin homolog (PTEN) (37%) are the most common mutations [26,28]. Thus, women with low-grade clear-cell EOC could benefit from treatment with mTor inhibitors targeting the PI3K/AKT/mTor pathway [21]. A good amount of research was also done in terms of targeted therapy for AT-rich Interactive Domain 1A (ARID1A) mutated clear-cell carcinoma, such as ENMD-2076 [29]. In the same way, small molecule inhibitors of the bromodomain and extra terminal domain (BET) family of proteins specifically inhibit proliferation of ARID1A mutated cell lines, both in vitro and in ovarian clear-cell cancer xenografts and patient-derived xenograft models. BET inhibitors cause a reduction in the expression of multiple SWI/SNF members including ARID1B, providing a potential explanation for the observed lethal interaction with ARID1A loss [30].

Endometrioid ovarian cancers and clear-cell ovarian cancers arise from similar precursor cells of transformed endometrial origin. Women with Lynch syndrome are also at an increased risk of developing endometrioid and clear-cell ovarian cancer. Loss of expression of mismatch repair genes (a characteristic of Lynch syndrome) is found in nearly 8% of Lynch syndrome-associated endometrioid and clear-cell ovarian cancers [31]. Molecular profiling of human endometrioid ovarian cancers revealed that the most prevalent mutations are similar to those observed in clear-cell ovarian cancer and include PIK3CA (40%), ARID1A (30%), KRAS (30%), PTEN (16%), and PPP2R1A (16%) [32,33]. Interestingly, mutations in CTNNB1 are particularly common, occurring in approximately 50% of low-grade endometrioid ovarian cancers [32,34]. High-grade endometrioid is a different entity than low-grade endometrioid and shares similarities with high grade serous EOC.

Mucinous ovarian carcinoma responds poorly to conventional chemotherapy regimens. Although long overall survival outcomes can occur with early detection and optimal surgical resection, recurrent and advanced disease are associated with extremely poor survival. Molecular aberrations noted in mucinous ovarian cancer that suggest a match with current targeted therapies include amplification of ERBB2 (26.7%) and BRAF mutation (9%). Observed genetic events that suggest potential efficacy for agents currently in clinical trials include: KRAS/NRAS mutation (66%), TP53 missense mutation (49%), RNF43 mutation (11%), ARID1A mutation (10%), and PIK3CA/PTEN mutation (9%). Therapies exploiting homologous recombination deficiency (see below) may not be effective in mucinous ovarian cancer, as only one out of 191 had a high homologous recombination deficiency score [23,35].

Nevertheless, it is not easy to evaluate the effectiveness of these new targeted therapies in the different histologic and molecular subtypes of EOC. Type 1 EOCs, for instance, which account for just 25% of all EOCs, are mostly diagnosed at an early stage when they are limited to the ovaries and can be treated surgically without the need for adjuvant therapy. Just 10% are diagnosed at a more advanced stage (carcinomatosis or lymph node involvement), which is when they need treatment with adjuvant systemic therapy. The scarcity of advanced cases thus makes it very difficult, if not impossible, to form a sufficiently large cohort to analyze the effectiveness of targeted treatments. That said, a number of trials have been conducted, but many of them are inconclusive because of lack of inclusion. For example, the phase III MILO trial (NCT01849874), designed to evaluate the efficacy of the MEK inhibitor binimetinib in low-grade serous EOCs, had to be stopped prematurely due to a lack of efficacy. Thus, drugs targeting signaling pathways need to be mindful that many of the trials have closed with poor results, suggesting that a dualistic approach to treatment is likely that way forward, particularly for mucinous cancers.

### 3.3. BRCAness: Definition and Treatment Implications

Between 30% and 40% of patients with type II EOC have a *BRCA* mutation. The *BRCA1* gene is located on chromosome 17. *BRCA1* mutations are of germline or somatic origin in 11.5% of cases and of epigenetic origin (inactivation due to gene promoter methylation) in 10.5%. The *BRCA2* gene is located on chromosome 14 and 9.2% of patients have germline or somatic mutations [36]. BRCA mutant carriers have a better prognosis than wild-type carriers or patients with epigenetic silencing [12]. *BRCA* genes are involved in DNA double-strand break repair by homologous recombination. When DNA damage occurs (induced, for example, by platinum-based chemotherapy), there are two possible repair pathways: the homologous recombination double-strand DNA repair pathway and the alternative single-strand break repair pathway, which involves a group of non-homologous end joining (*NHEJ*) genes and single-strand annealing (*SSA*) genes that use several enzymes, including PARP (Figure 1). In the absence of double-strand break repair due to, for example, deficient *BRCA1* or *BRCA2* function, the default single-strand repair pathway is engaged, but this results in genomic instability in the reading frame and genomic mutations. This instability would explain the higher prevalence of EOC in patients with *BRCA1* or *BRCA2* mutations, as there would be an accumulation of cell clones harboring activating mutations in oncogenes or inhibiting mutations in tumor suppressor genes. Defective DNA repair at the tumor level exists in EOC, as 30% to 40% of patients have a non-functional *BRCA* gene. This deficiency probably explains the better prognosis observed in carriers of *BRCA* mutations when their tumor is subjected to selective pressure imposed by platinum-based chemotherapy, as the single-strand repair capacity would probably be exceeded. Accumulation of unrepaired DNA damage results in cell apoptosis once a certain threshold is passed. PARP inhibitors such as olaparib can be used to exploit double-strand DNA repair deficiencies by inhibiting the single-strand DNA repair pathway and preventing DNA repair in patients treated with platinum-based chemotherapy. In the presence of cumulative unrepaired DNA damage, the tumor cells would enter apoptosis. This process is known as synthetic lethality. Synthetic lethality induced by PARP inhibitors has been found to result in a significant clinical benefit in patients with ovarian cancer. Specifically, carriers of constitutional (inherited) or somatic (acquired) *BRCA* mutations achieved a significant gain of seven months disease-free survival in patients treated with olaparib maintenance therapy for at least nine months [37]. A number of questions, however, remain in relation to the use of PARP inhibitors. First, would PARP inhibitors benefit patients who lack *BRCA* expression due to epigenetic inactivation through DNA methylation? Second, would all EOC patients with a BRCAness profile benefit from PARP inhibitors, i.e., patients with a mutation in any of the proteins responsible for double-strand DNA repair (homologous recombination) whether that be BRCA or BRIP2, CHEK2, RAD51c, or PTEN [38,39]? Third, can the somatic characterization of the BRCAness profile be simplified in order to offer theranostic options to patients other than those with congenital *BRCA* mutations? Although the definition of BRCAness varies from one series to the next, molecular profiling of high-grade serous EOCs has shown that 30% to 60% of patients have deficiencies in homologous recombination or other DNA repair pathways [14,38,40]. A number of studies have attempted to identify molecular signatures that would provide a simple, reproducible method for predicting BRCAness profiles based on genomic instability [41,42]. Similar studies have been conducted in the field of breast cancer, particularly triple-negative breast cancer [43,44]. The difficulty with these studies is the lack of consensus on the molecular definition of BRCAness. The strategy proposed for overcoming this difficulty in therapeutic trials is to use PARP inhibitors for patients who respond well to platinum-based drugs and therefore theoretically have defective double-strand DNA repair. Two examples are the phase II ARIEL3 trial with rucaparib and the NOVA trial with niraparib. The NOVA trial showed that niraparib was effective in patients with platinum-sensitive ovarian cancer, demonstrating that PARP inhibitors are effective in carriers of mutations other than *BRCA1* and *BRCA2* [24]. Efforts are also underway to analyze the combined use of drugs with different effects. Researchers in the PAOLA trial, for example, recently showed that the combined use of PARP inhibitors and antiangiogenic agents as first-line maintenance therapy for patients with ovarian cancer is superior to bevacizumab alone [45,46].

### 3.4. Four Subtypes of Serous Papillary EOC: Diagnostic and Therapeutic Implications

Large-scale molecular characterization of several hundred high-grade serous ovarian cancers from the Cancer Genome Atlas Research Network led to the identification of four molecular subtypes of ovarian cancer (Table 3): (1) a mesenchymal subtype with gene expression in the stromal component (e.g., fibroblasts, vascular pericytes), (2) a proliferative subtype with high expression of transcription factors and proliferation markers, (3) a differentiated subtype with expression of differentiation markers, and (4) an immunoreactive subtype with high expression of T-cell chemokine ligands, major histocompatibility complex I and II genes, and PD-L1 (programmed cell death ligand 1) genes (Table 2) [14]. This gene expression–based signature was created using 1500 genes and then simplified to create a prognostic “Classification of Ovarian Cancer” (CLOVAR) model based on 100 genes [47]. According to the CLOVAR signature, the mesenchymal subtype (19% of cases) is associated with the least favorable prognosis, with a five-year survival rate of just 18%. This is followed by the proliferative subtype (25% of cases) with a five-year survival of 22%, the differentiated subtype (25% of cases) with a five-year survival of 40%, and the immunoreactive subtype with a five-year survival of 42% [40]. The CLOVAR model can be used to distinguish between different types of high-grade serous papillary ovarian cancers and could have important theranostic implications. In fact, a retrospective analysis of data from the ICON7 clinical trial showed that bevacizumab provided a survival benefit bordering on significance (*p* = 0.053) in patients with the mesenchymal subtype of serous ovarian cancer; the survival gain observed for the other subtypes was clearly non-significant [48]. Likewise, patients with the immunoreactive subtype, characterized by high expression of Programmed Death-ligand 1 (PD-L1), could potentially benefit from anti-PD1 and anti-PD-L1 monoclonal antibodies. These new immunotherapy drugs can reverse tumor-mediated immunosuppression, providing a survival advantage that has already been observed in melanoma and lung cancer [49]. In ovarian cancer, phase II [50] and ongoing phase III trials have also reported very encouraging results. PD1 and PD-L1 inhibitors may also be an option for maintenance therapy, although numerous issues remain to be investigated, such as societal costs and treatment duration. The prognostic CLOVAR signature is still at the research stage but could have valuable applications in clinical practice in the future. One of the shortcomings of this model is that high-grade serous papillary EOC subtypes are not mutually exclusive, with one study showing that 40% of cancers could be assigned to two subtypes [40]. Breast cancer is different in this respect, as subtypes such as luminal A, luminal B, and triple-negative breast cancer are mutually exclusive. This mutual exclusion enables a clear classification system that is very useful for stratifying systemic treatments.

In another way, compared with high grade serous ovarian cancer, most clear-cell ovarian cancers are characterized by considerably fewer copy number alterations, and mutations in TP53 and BRCA1/BRCA2 genes are uncommon [51]. However, recently, a pilot study highlighted a unique subset of clear-cell ovarian cancer that have microsatellite instability associated with enhanced immunogenicity, which may be susceptible to immunotherapy. This subgroup of clear-cell ovarian cancer exhibited a significantly higher number of tumor-infiltrating lymphocytes (TILs), particularly programmed cell death protein 1 (PD-1)-positive TILs, compared with microsatellite-stable clear-cell ovarian cancer as well as high grade serous ovarian cancer, and uniformly expressed programmed death-ligand 1 (PD-L1) in tumor cells and/or intraepithelial or peritumoral immune cells [52]. Thus, this clear-cell ovarian cancer subtype could potentially benefit from anti-PD1 and anti-PD-L1 monoclonal antibodies.

## 4. Conclusions

EOC is an aggressive cancer and type II EOC has a particularly poor prognosis (overall five-year survival rate <20%) as it is normally diagnosed at an advanced stage when peritoneal carcinomatosis has already occurred. Molecular characterization of EOC has already led to the identification of extremely interesting treatments, such as PARP inhibitors for patients with a BRCAness profile and immunotherapy drugs capable of reverting tumor-induced immunosuppression. These new drugs offer hope in a field that has seen no significant improvements in survival for several decades. Much work, however, remains to improve the molecular characterization of EOC and pave the way for targeted therapies and long-awaited personalized medicine for women with ovarian cancer. Finally, the use of PARP inhibitors, PD1/PD-L1 inhibitors, and antiangiogenic drugs such as bevacizumab as maintenance therapy is likely to change current treatment paradigms. The goal of the current standard of care—surgery and systemic chemotherapy—is to eliminate or reduce tumor burden to a minimum, but the extent of residual disease varies and maintenance therapy could offer a means of controlling this “ever-present” disease. The definitions of platinum-sensitive and -resistant EOC based on time of disease recurrence (before or after six months of the end of chemotherapy) that currently determine the need for platinum retreatment will probably evolve as new treatments result in longer disease-free periods. Similarly, the concept of late recurrence (i.e., at least 18 months disease free), which could be addressed by new surgery if localized, will probably need to be reviewed. Finally, the increasing use of maintenance treatments capable of slowing disease progression will probably mean that the current surgical dogma of achieving zero residual disease will probably no longer apply in a few decades.

## Figures and Tables

**Figure 1 jcm-09-02239-f001:**
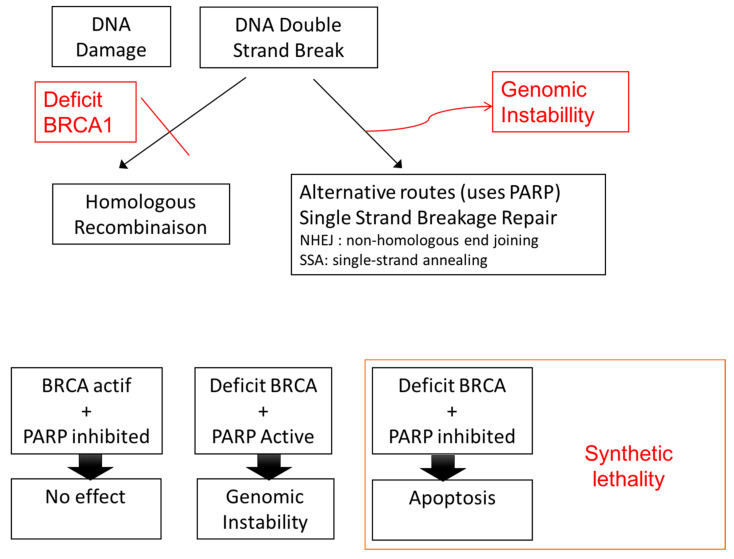
Concept of synthetic lethality. PARP: Poly Adenosine Diphosphate-Ribose Polymerase; BRCA: Breast Cancer.

**Table 1 jcm-09-02239-t001:** Dualistic classification of EOC.

	Type I EOC	Type II EOC
Frequency	25% of EOCs	75% of EOCs
Mortality	10%	90%
Diagnosis	Usually Early stage	Usually Advanced stage (carcinomatosis)
Progression	Slow	Fast
Histologic subtypes	Low-grade serous Low-grade endometrioid Clear-cell Mucinous	High-grade serous High-grade endometrioid Undifferentiated carcinoma Carcinosarcoma
Mutations	*KRAS*, *BRAF* (serous) *CTTNNB1*, *PTEN*, *PI3KCA* (endometrioid) *KRAS*, *BRAF*, *HER2*, *P53* (mucinous) (P53 for 25% of mucinous EOC) *PI3KCA*, *HER2* (clear-cell) *ARID1A*, *PPP2R1A*	*P53* (96% of high-grade serous EOCs) *BRCA* (21% of cases)
Genomic stability	Stability	Instability
Precursors	Benigh cyst » borderline » malignant	Precursors: de novo, STIC

Abbreviations: EOC: epithelial ovarian cancer; STIC: serous tubal intraepithelial carcinoma.

**Table 2 jcm-09-02239-t002:** EOC classified according Histologic Subtype.

Histologic Subtype	High-Grade Serous	Low-Grade Serous	Mucinous	Endometrioid	Clear-Cell
% of all EOCs diagnosed	71%	4%	3%	8%	9%
% of early stage EOCs diagnosed	Nearly 0%	17%	12%	33%	38%
% of advanced stage EOCs diagnosed	88%	3%	2%	3%	4%
Progression speed	High	low	low (50%) high (50%)	Low (90%) High (10%)	low
Mutations	*P53* *BRCA1* or *BRCA2*	*KRAS* *BRAS* *HER2*	*P53* *KRAS*	*PTEN* *PI3KCA*	*PI3KCA*
Intracellular signaling pathway targeted	P53, which controls mitosis and apoptosis	MEK/BRAF/KRAS pathway		PI3K/AKT/mTor	
Function of signaling pathway	BRCA involved in double-strand DNA repair	Cell proliferation		Regulates cell proliferation, motility, and survival	
Drugs targeting signaling pathway	PARP inhibitor (olaparib) for tumors with a BRCAness profile	Selumetinib, trametinib, dabrafenib		mTor inhibitor (e.g., temsirolimus, everolimus)	

Abbreviations: EOC: epithelial ovarian cancer; BRCA: Breast Cancer; PARP: Poly Adenosine Diphosphate-Ribose Polymerase; MEK: mitogen-activated protein kinase kinase enzymes; BRAF: B-Raf protein; KRAS: K-ras Oncogene; PI3K: phosphatidylinositol-4,5-biphosphate 3-kinase; AKT: Activation Of The Serine/Threonine-specific Protein Kinase; mTor: Mammalian Target Of Rapamycin; P53: Protein 53.

**Table 3 jcm-09-02239-t003:** Four subtypes of high-grade serious ovarian cancer according to expression profile.

Subtypes of High-Grade Serous Ovarian Cancer	High Expression	Low Expression	Five-Year Survival	Potential Targeted Therapy
Mesenchymal	Genes in stromal component (fibroblasts, vascular pericytes, etc.)		18%	Bevacizumab (anti-VEGF)
Proliferative	Transcription factors and proliferation markers	Differentiation markers (MUC16 and MUC1)	22%	Bevacizumab (anti-VEGF)
Differentiated	Differentiation markers (MUC16 and MUC1)		42%	
Immunoreactive	T-cell chemokines MHC class I and II PD-L1		45%	Nivolumab (Anti PD1/PD-L1 monoclonal antibody)

Abbreviations: MHC: major histocompatibility complex; PD-L1: programmed cell death ligand 1; MUC: Mucin; VEGF: Vascular Endothelial Growth Factor.

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
