# Peer review of "The Landscape and Therapeutic Implications of Molecular Profiles in Epithelial Ovarian Cancer"

_jcm, 2020, doi:10.3390/jcm9072239_

Round 1

Reviewer 1 Report

Although the writing is clear, this review suffers from a lack of new information and synthetic views/insights generated by the authors. The contents are a mere summary of what we have known so far and already covered by other review articles.

Author Response

Authors thank reviewer for their comments. See below answer for queries.

1°) The draft was edited by a native English, named Felicity Neilson, who worked in an editing company.

2°) Although the writing is clear, this review suffers from a lack of new information and synthetic views/insights generated by the authors. The contents are a mere summary of what we have known so far and already covered by other review articles.

Authors agree with reviewer that present review is a summary of concept and insights about what we know in ovarian cancer. Indeed, the principle of a review is about what is published in literature in order to do a state of art and propose perspective. Could reviewer indicate the information that is lacking? Thus, authors could add this information in a revised version of present work.

Reviewer 2 Report

This manuscript presents a short review of the molecular landscape and therapeutic Implications of all EOC subtypes, and a nice discussion and visual representation of BRCAness in EOC and present treatment implications

Major revision:

Whilst the dualistic classification of EOC into “type I” and “type II” is widely applied in research presentations and manuscripts, it is often used as a convenient way of conceptualizing different mechanisms of tumorigenesis amongst EOC. However, this classic dualistic classification conflicts with recent molecular studies as not all type I EOCs are not homogenous, even within the histological types, and a proportion can have poor clinical outcomes such as high grade mucinous. Type II serous and type I low grade serous carcinomas best fit the description of the dualistic model, with different precursors, and distinct very distinct pathology and molecular profiles. The very clear differences between mucinous ovarian carcinomas and other type I tumours, including an unknown cell of origin, and a very heterogeneous mutation profile (including TP53 mutant cases) and clinical behaviour, indicate a non-type I classification for this entity. The impression that only type II carcinomas are aggressive, have poor prognosis, and carry TP53 mutations can be unhelpful. This review should address this in the discussion

Minor revisions

Table 1 – It would be helpful to represent the frequency of the characteristics listed in each subtype in Table 1. Eg early stage diagnosis for Low-grade serous, Low-grade endometrioid and Mucinous are seen in greater than 50% of cases, whilst Clear-cell is less than 50% and perhaps closer to 10%. Additionally whilst progression is slow (<10% of cases) for low-grade, endo and clear cell, >50 % of mucinous carcinomas have a high proliferation rate and disease progression

Table 1 - Low-grade serous are not characterised by HER2, but a proportion of clear cell and mucinous are – need to adjust table

Table 1 - Mucinous are also BRAF, TP53 (in >25% of cases and almost 100% of high grade mucinous tumours) mutation positive – reference Gorringe et al, 2020 - Therapeutic options for mucinous ovarian carcinoma

The MEKi Selumetinib has shown little activity in patients with low-grade serous, where response did not always correlate with KRAS/BRAF mutation status. Indeed, clinical response to the BRAF inhibitor, dabrafenib, has been demonstrated thus far clinically in 20% women with BRAFV600E mutated low-grade carcinomas. The sentence “Women with low-grade, clear-cell, or endometrioid EOC, for example, could benefit from treatment with mTor inhibitors targeting the PI3K/AKT/mTor pathway” needs to adjusted and table 2 where is says Drugs targeting signalling pathway needs to be mindful that many of the trials have closed with poor results, suggesting that a dualistic approach to treatment is likely that way forward – particularly for mucinous. Take a look at references below and consider including

  1. Farley, J., et al., Selumetinib in women with recurrent low-grade serous carcinoma of the ovary or peritoneum: an open-label, single-arm, phase 2 study. Lancet Oncol, 2013. 14(2): p. 134-40.
  2. Moujaber, T., et al., BRAF Mutations in Low-Grade Serous Ovarian Cancer and Response to BRAF Inhibition. JCO Precision Oncology, 2018(2): p. 1-14.

Table 2 does not have a title

Author Response

Authors thank reviewer for their clever comments that allowed to improve manuscript. See below answer for queries.

1°) The draft was edited by a native English, named Felicity Neilson, who worked in an editing company.

2°) Whilst the dualistic classification of EOC into “type I” and “type II” is widely applied in research presentations and manuscripts, it is often used as a convenient way of conceptualizing different mechanisms of tumorigenesis amongst EOC. However, this classic dualistic classification conflicts with recent molecular studies as not all type I EOCs are not homogenous, even within the histological types, and a proportion can have poor clinical outcomes such as high grade mucinous. Type II serous and type I low grade serous carcinomas best fit the description of the dualistic model, with different precursors, and distinct very distinct pathology and molecular profiles. The very clear differences between mucinous ovarian carcinomas and other type I tumours, including an unknown cell of origin, and a very heterogeneous mutation profile (including TP53 mutant cases) and clinical behaviour, indicate a non-type I classification for this entity.

Authors do totally agree with this comment. The followed sentences were added in page 6 of present work. “Nevertheless, whilst the dualistic classification of EOC into “type I” and “type II” is widely applied in research presentations and manuscripts, it is often used as a convenient way of conceptualizing different mechanisms of tumorigenesis amongst EOC. This classic dualistic classification conflicts with some recent molecular studies as not all type I EOCs are not homogenous, even within the histological types, and a proportion can have poor clinical outcomes such as high grade mucinous. Indeed, the impression that only type II carcinomas are aggressive, have poor prognosis, and carry TP53 mutations can be unhelpful. Although, type II serous and type I low grade serous carcinomas best fit the description of the dualistic model, with different precursors, and distinct very distinct pathology and molecular profiles., there are very clear differences between mucinous ovarian carcinomas and other type I tumours, including an unknown cell of origin, and a very heterogeneous mutation profile (including TP53 mutant cases) and clinical behaviour, indicate a non-type I classification for this entity. Mucinous ovarian carcinoma responds poorly to conventional chemotherapy regimens. Although long overall survival outcomes can occur with early detection and optimal surgical resection, recurrent and advanced disease are associated with extremely poor survival. Molecular aberrations noted in mucinous ovarian cancer that suggest a match with current targeted therapies include amplification of ERBB2 (26.7%) and BRAF mutation (9%). Observed genetic events that suggest potential efficacy for agents currently in clinical trials include: KRAS/NRAS mutations (66%), TP53 missense mutation (49%), RNF43 mutation (11%), ARID1A mutation (10%), and PIK3CA/PTEN mutation (9%). Therapies exploiting homologous recombination deficiency (see below) may not be effective in MOC, as only 1/191 had a high homologous recombination deficiency score (1Gorringe KL, Cheasley D, Wakefield MJ, et al. Therapeutic Options for mucinous ovarian carcinoma. Gynecol Oncol 2020;156:552-60.)”

3°) Table 1 – It would be helpful to represent the frequency of the characteristics listed in each subtype in Table 1. Eg early stage diagnosis for Low-grade serous, Low-grade endometrioid and Mucinous are seen in greater than 50% of cases, whilst Clear-cell is less than 50% and perhaps closer to 10%. Additionally whilst progression is slow (<10% of cases) for low-grade, endo and clear cell, >50 % of mucinous carcinomas have a high proliferation rate and disease progression

Authors agree with this point of view, but, with this new information, the presentation is difficult for table 1. Thus, we added this information in table 2. See highlighted adding in table 2.

4°) Table 1 - Low-grade serous are not characterised by HER2, but a proportion of clear cell and mucinous are – need to adjust table

As request, the table 1 was amended.

5°) Table 1 - Mucinous are also BRAF, TP53 (in >25% of cases and almost 100% of high grade mucinous tumours) mutation positive – reference Gorringe et al, 2020 - Therapeutic options for mucinous ovarian carcinoma

As request, the table 1 was amended.

6°) The MEKi Selumetinib has shown little activity in patients with low-grade serous, where response did not always correlate with KRAS/BRAF mutation status. Indeed, clinical response to the BRAF inhibitor, dabrafenib, has been demonstrated thus far clinically in 20% women with BRAFV600E mutated low-grade carcinomas. The sentence “Women with low-grade, clear-cell, or endometrioid EOC, for example, could benefit from treatment with mTor inhibitors targeting the PI3K/AKT/mTor pathway” needs to adjusted and table 2 where is says Drugs targeting signalling pathway needs to be mindful that many of the trials have closed with poor results, suggesting that a dualistic approach to treatment is likely that way forward – particularly for mucinous. Take a look at references below and consider including

  1. Farley, J., et al., Selumetinib in women with recurrent low-grade serous carcinoma of the ovary or peritoneum: an open-label, single-arm, phase 2 study. Lancet Oncol, 2013. 14(2): p. 134-40.
  2. Moujaber, T., et al., BRAF Mutations in Low-Grade Serous Ovarian Cancer and Response to BRAF Inhibition. JCO Precision Oncology, 2018(2): p. 1-14.

Authors agree with this comment. Page 6, the following sentences were added: “That said, a number of trials have been conducted, but many of them are inconclusive because of lack of inclusion. Nevertheless, some of them had to be conducted, but with poor results.  For example, the phase III MILO trial (NCT01849874), for example, designed to evaluate the efficacy of the MEK inhibitor binimetinib in low-grade serous EOCs, had to be stopped prematurely due to a lack of efficacy. In the same way, the MEKi Selumetinib has shown little activity in patients with low-grade serous (15%), where response did not always correlate with KRAS/BRAF mutation status. With another MEKi, clinical response to the BRAF inhibitor, dabrafenib, has been demonstrated thus far clinically in 20% women with BRAFV600E mutated low-grade carcinomas 2. Thus, drugs targeting signaling pathway needs to be mindful that many of the trials have closed with poor results, suggesting that a dualistic approach to treatment is likely that way forward – particularly for mucinous.”

7°) Table 2 does not have a title

Reviewer is right. The followed title was added: “EOC classified according Histologic Subtype”

Round 2

Reviewer 1 Report

This revised version shows some improvement from the original version. However, this manuscript is still suffered by limited discussion on the recent findings and most of the references are old (only 5 out of 33 references are published in the last 3 years). For example, a good amount of research was done in terms of targeted therapy for ARI1A mutated clear cell carcinoma and some serous subtype (e.g. FAK inhibitor for stromal/mesenchymal subtype) but this was not discussed at all in the review. Also, as the authors stated, type I and II EOC classification is criticized as an outdated model. The subtype-specific genetic mutations they discussed are also outdated (as they referred a review article in 2013) and need to be updated. It is not clear why the authors did not include the mutations of type I EOC (e.g. CTNNB1 for endometrioid; BRAF for mucinous; ARID1A for clear cell) they mentioned in Table 2. If the authors cited review articles, please make sure to refer to the most recent/updated review articles on the topic.

It would be beneficial to add ‘Drugs targeting signaling pathways’ to Table 3 as they did in Table 2.

More thorough editing is required since there are several errors in the text and table (e.g. BRCR1/2 (in p2) à BRCA1/2; delete ‘t Grade)’ from type II EOC section in Table 1; BRCA1 or BRCA1 mutation (p6 line 11) à BRCA1 or BRCA2 mutation; fix reference errors in p6 lines 37 and 41 and p7 line 68; incorporate newly added 3 references  into the Reference section).

Author Response

Authors thank reviewer for their comments in order to improve present work. See below answers to questions.

1°) This revised version shows some improvement from the original version. However, this manuscript is still suffered by limited discussion on the recent findings and most of the references are old (only 5 out of 33 references are published in the last 3 years).

As requested, we added many recent references were added in this revised draft.

See below, the references added:  

  • Lheureux S, Braunstein M, Oza AM. Epithelial ovarian cancer: Evolution of management in the era of precision medicine. CA Cancer J Clin 2019;69:280-304.
  • Kroeger PTJr, Drapkin R. Pathogenesisand heterogeneity of ovarian cancer.Curr Opin Obstet Gynecol. 2017;29:26-34.
  • Pauly N, Ehmann S, Ricciardi E, Ataseven B, Bommert M, Heitz F, Prader S, Schneider S, du Bois A, Harter P, Baert T.Low-grade Serous Tumors: Are We Making Progress?Curr Oncol Rep. 2020 Jan 27;22(1):8. doi: 10.1007/s11912-020-0872-5
  • Itamochi H, Oishi T, Oumi N, et al. Whole genome sequencing revealed novel prognostic biomarkers and promising targets for therapy of ovarian clear cell carcinoma. Br J Cancer. 2017;117:717-724., -----
  • Shibuya Y, Tokunaga H, Saito S, Shimokawa K, Katsuoka F, Bin L, et al. Identification of somatic genetic alterations in ovarian clear cell carcinoma with next generation sequencing. Genes Chromosomes Cancer. 2018;57:51-60.

-       Marks EI, Brown VS, Dizon DS.Genomic and Molecular Abnormalities in Gynecologic Clear Cell Carcinoma.Am J Clin Oncol. 2020 Feb;43(2):139-145.

-       Fukumoto T, Magno E, Zhang R.SWI/SNF Complexes in Ovarian Cancer: Mechanistic Insights and Therapeutic Implications.Mol Cancer Res. 2018 Dec;16(12):1819-1825. doi: 10.1158/1541-7786.MCR-18-0368. Epub 2018 Jul 23.

-       Stephanie Lheureux 1, Anna Tinker 2, Blaise Clarke 3, Prafull Ghatage 4, Stephen Welch 5, Johanne I Weberpals 6, Neesha C Dhani 1, Marcus O Butler 1, Katia Tonkin 7, Qian Tan 1, David S P Tan 8, Kelly Brooks 1, Janelle Ramsahai 1, Lisa Wang 9, Nhu-An Pham 3, Patricia A Shaw 3, Ming S Tsao 3, Swati Garg 10, Tracey Stockley 10 11 12, Amit M Oza  Clin Cancer research 2018 Dec 15;24(24):6168-6174. doi: 10.1158/1078-0432.CCR-18-1244. Epub 2018 Aug 14. A Clinical and Molecular Phase II Trial of Oral ENMD-2076 in Ovarian Clear Cell Carcinoma (OCCC): A Study of the Princess Margaret Phase II Consortium

-       Berns K, Caumanns JJ, Hijmans EM, Gennissen AMC, Severson TM, Evers B, Wisman GBA, Jan Meersma G, Lieftink C, Beijersbergen RL, Itamochi H, van der Zee AGJ, de Jong S, Bernards R ARID1A mutation sensitizes most ovarian clear cell carcinomas to BET inhibitors..Oncogene. 2018 Aug;37(33):4611-4625. doi: 10.1038/s41388-018-0300-6. Epub 2018 May 15.PMID: 29760405 )

2°) For example, a good amount of research was done in terms of targeted therapy for ARI1A mutated clear cell carcinoma and some serous subtype (e.g. FAK inhibitor for stromal/mesenchymal subtype) but this was not discussed at all in the review.

As request, some research that targeted therapy for ARID1A mutated clear cell carcinoma were added.

The followed sentences were added: A good amount of research was also done in terms of targeted therapy for ARID1A mutated clear cell carcinoma, such as ENMD-2076. (Clin Cancer research 2018 Dec 15;24(24):6168-6174. doi: 10.1158/1078-0432.CCR-18-1244. Epub 2018 Aug 14. A Clinical and Molecular Phase II Trial of Oral ENMD-2076 in Ovarian Clear Cell Carcinoma (OCCC): A Study of the Princess Margaret Phase II Consortium Stephanie Lheureux 1, Anna Tinker 2, Blaise Clarke 3, Prafull Ghatage 4, Stephen Welch 5, Johanne I Weberpals 6, Neesha C Dhani 1, Marcus O Butler 1, Katia Tonkin 7, Qian Tan 1, David S P Tan 8, Kelly Brooks 1, Janelle Ramsahai 1, Lisa Wang 9, Nhu-An Pham 3, Patricia A Shaw 3, Ming S Tsao 3, Swati Garg 10, Tracey Stockley 10 11 12, Amit M Oza) 

In the same way, small molecule inhibitors of the BET (bromodomain and extra terminal domain) family of proteins specifically inhibit proliferation of ARID1A mutated cell lines, both in vitro and in ovarian clear cell cancer xenografts and patient-derived xenograft models. BET inhibitors cause a reduction in the expression of multiple SWI/SNF members including ARID1B, providing a potential explanation for the observed lethal interaction with ARID1A loss.  (ARID1A mutation sensitizes most ovarian clear cell carcinomas to BET inhibitors.Berns K, Caumanns JJ, Hijmans EM, Gennissen AMC, Severson TM, Evers B, Wisman GBA, Jan Meersma G, Lieftink C, Beijersbergen RL, Itamochi H, van der Zee AGJ, de Jong S, Bernards R.Oncogene. 2018 Aug;37(33):4611-4625. doi: 10.1038/s41388-018-0300-6. Epub 2018 May 15.PMID: 29760405 )

3°) Also, as the authors stated, type I and II EOC classification is criticized as an outdated model.

Authors agree with point and authors stated this in present review. Authors would present the evolution of model from type I and 2 EOC classification (showed in table 1) to the five sub-classifications of ovarian cancer (high grade serous ovarian cancer, low grade ovarian cancer, clear cell ovarian cancer, mucinous ovarian cancer, low grade endometrioid ovarian cancer showed in table 2).

4°) The subtype-specific genetic mutations they discussed are also outdated (as they referred a review article in 2013) and need to be updated.

The subtype specific genetic mutations were adapted with recent reviews and recent published data such as:

  • Lheureux S, Braunstein M, Oza AM. Epithelial ovarian cancer: Evolution of management in the era of precision medicine. CA Cancer J Clin 2019;69:280-304.

-       Marks EI, Brown VS, Dizon DS Genomic and Molecular Abnormalities in Gynecologic Clear Cell Carcinoma..Am J Clin Oncol. 2020 Feb;43(2):139-145.

  • Low-grade Serous Tumors: Are We Making Progress?Pauly N, Ehmann S, Ricciardi E, Ataseven B, Bommert M, Heitz F, Prader S, Schneider S, du Bois A, Harter P, Baert T.Curr Oncol Rep. 2020 Jan 27;22(1):8. doi: 10.1007/s11912-020-0872-5

5°) It is not clear why the authors did not include the mutations of type I EOC (e.g. CTNNB1 for endometrioid; BRAF for mucinous; ARID1A for clear cell) they mentioned in Table 2.

These mutations (CTNNB1 for endometrioid; BRAF for mucinous; ARID1A for clear cell) mentioned in table 2 were added and discussed in the review.

6°) If the authors cited review articles, please make sure to refer to the most recent/updated review articles on the topic.

This point was done (see answer question 1).

7°) It would be beneficial to add ‘Drugs targeting signaling pathways’ to Table 3 as they did in Table 2.

A column with potential targeted therapy was added in table 3.

8°) More thorough editing is required since there are several errors in the text and table (e.g. BRCR1/2 (in p2) à BRCA1/2;

This mistake was corrected.

9°) delete ‘t Grade)’ from type II EOC section in Table 1;

This mistake was corrected.

10°) BRCA1 or BRCA1 mutation (p6 line 11) à BRCA1 or BRCA2 mutation;

This mistake was corrected.

11°) fix reference errors in p6 lines 37 and 41 and p7 line 68; incorporate newly added 3 references into the Reference section

The references were added in reference section.

Reviewer 2 Report

The authors will need to rephrase their answers to question 2 and 6, as they have copied what I said and put it in the text

Author Response

Authors modified the text in relation to answer of questions 2 and 6. 

Round 3

Reviewer 1 Report

The authors have addressed the reviewer's comments and improved the manuscript.

Please fix a few minor errors in the text (e.g. p6 line 50, KRAS) -> KRAS; p7 line 78, dysplays mutations in mutations in-> displays mutations in) before publication. 

Author Response

Question : Please fix a few minor errors in the text (e.g. p6 line 50, KRAS) -> KRAS; p7 line 78, dysplays mutations in mutations in-> displays mutations in) before publication. 

Response: 

Authors perform corrections of misspelling.